# Multiple Lines of Evidences Reveal Mechanisms Underpinning Mercury Resistance and Volatilization by *Stenotrophomonas* sp. MA5 Isolated from the Savannah River Site (SRS), USA

**DOI:** 10.3390/cells8040309

**Published:** 2019-04-03

**Authors:** Meenakshi Agarwal, Rajesh Singh Rathore, Charles Jagoe, Ashvini Chauhan

**Affiliations:** 1Environmental Biotechnology Laboratory, School of the Environment, 1515 S. Martin Luther King Jr. Blvd., Suite 305B, FSH Science Research Center, Florida A&M University, Tallahassee, FL 32307, USA; meenakshiagarwal.iitb@gmail.com (M.A.); rajeshrathore854@gmail.com (R.S.R.); 2Environmental Toxicology Laboratory, School of the Environment, 1515 S. Martin Luther King Jr. Blvd., Suite 305B, FSH Science Research Center, Florida A&M University, Tallahassee, FL 32307, USA; chjagoe@gmail.com

**Keywords:** mercury, *Stenotrophomonas* sp., *mer* operon, whole genome sequence analysis

## Abstract

A largely understudied microbially mediated mercury (Hg) bioremediative pathway includes the volatilization of Hg^2+^ to Hg^0^. Therefore, studies on Hg resistant bacteria (HgR), isolated from historically long-term contaminated environments, can serve as models to understand mechanisms underpinning Hg cycling. Towards this end, a mercury resistant bacterial strain, identified as *Stenotrophomonas* sp., strain MA5, was isolated from Mill Branch on the Savannah River Site (SRS); an Hg-impacted ecosystem. Minimum inhibitory concentration (MIC) analysis showed Hg resistance of up to 20 µg/mL by MA5 with 95% of cells retaining viability. Microcosm studies showed that the strain depleted more than 90% of spiked Hg^2+^ within the first 24 h of growth and the detection of volatilized mercury indicated that the strain was able to reduce Hg^2+^ to Hg^0^. To understand molecular mechanisms of Hg volatilization, a draft whole genome sequence was obtained, annotated and analyzed, which revealed the presence of a transposon-derived *mer* operon (*merRTPADE*) in MA5, known to transport and reduce Hg^2+^ into Hg^0^. Based on the whole genome sequence of strain MA5, qRT-PCR assays were designed on *merRTPADE*, we found a ~40-fold higher transcription of *mer T, P, A, D* and *E* when cells were exposed to 5 µg/mL Hg^2+^. Interestingly, strain MA5 increased cellular size as a function of increasing Hg concentrations, which is likely an evolutionary response mechanism to cope with Hg stress. Moreover, metal contaminated environments are shown to co-select for antibiotic resistance. When MA5 was screened for antibiotic resistance, broad resistance against penicillin, streptomycin, tetracycline, ampicillin, rifampicin, and erythromycin was found; this correlated with the presence of multiple gene determinants for antibiotic resistance within the whole genome sequence of MA5. Overall, this study provides an in-depth understanding of the underpinnings of *Stenotrophomonas*-mercury interactions that facilitate cellular survival in a contaminated soil habitat.

## 1. Introduction

The U.S. Department of Energy (DoE) managed Savannah River Site (SRS), located in South Carolina (SC), is a former nuclear production facility where mercury (Hg) contamination from past industrial and nuclear production activities is still prevalent [1]. SRS received Hg-containing discharges from industrial activities from the 1960s to 1980s [2]. Hg is a mobile heavy metal and therefore, the environmental fate, dictated mostly by microbially-mediated transformations, includes transformations among elemental Hg (Hg^0^), oxidized divalent (Hg^2+^) and neurotoxic methylated Hg, with the latter posing the most ecological and human health concerns due to its propensity to biomagnify through food webs [3,4]. Overall, both divalent and methyl Hg are well documented to exert toxicity to ecosystem health and processes [3,5,6].

Microbes have evolved various adaptive mechanisms to survive in Hg-contaminated environments [7]. One such mechanism is governed by the *mer* operon which can be located on plasmid(s), chromosome(s) or may even be a component of transposons or integrons [8,9,10]. In general, the *mer* operon consists of regulatory proteins, such as the MerR and MerD, transport proteins like MerP, T and E, as well as protein(s) with reductase activity like MerA [11,12]. The MerR protein binds within the promoter-operator region and regulates the transcription of the structural gene depending on Hg availability, or lack thereof. MerD also binds within the same promoter-operator region but probably with weak interaction whereas, proteins MerP and MerT aid in the transport of mercuric ions from the periplasm to the cytoplasm within the bacterial cell, making it available to MerA, which codes for mercuric reductase, an NAD(P)H dependent flavin oxidoreductase. Mercuric reductase acts to reduce Hg^2+^ into volatile Hg^0^, which then passively diffuses out from the cellular environment. Depending upon the *mer* determinants, mercury resistance has been classified into broad or narrow spectrums; the broad spectrum mercury resistant bacteria carry the *merB* gene, in addition to the *merA,* which codes for an organomercurial lyase, and converts methylmercury into Hg^2+^ via a demethylation process which further reduces into Hg^0^ by the action of MerA [11,13]. Thus, MerA performs a central role in both, narrow as well as broad-spectrum Hg resistant (HgR) bacteria in the environment.

Given this background, HgR microbiota can be targeted for remediation of Hg but firstly, isolation and evaluation of the molecular and cellular mechanisms underpinning Hg resistance need to be carefully examined, which can vary significantly as a function of the environment being investigated. Towards this end, the SRS site, with a long-term history of exposure to Hg, provides an opportunity to isolate and evaluate HgR microbiota and their bioremediative mechanisms. As part of a larger study, we have obtained a plethora of soil-borne bacterial strains exhibiting variable Hg resistance abilities. This study reports the genomic, physiological and cellular characterization of a robust Hg resistant strain, identified as *Stenotrophomonas* sp., strain MA5. It is noteworthy that *Stenotrophomonas maltophilia*, previously known as *Pseudomonas maltophilia,* is the third most common nosocomial opportunistic pathogen [14], belonging to the Xanthomonadaceae family. *Stenotrophomonas* sp., are ubiquitously distributed both, in the environment and in clinical settings, exhibiting multidrug-resistance associated with bone, joint, respiratory, and urinary tract infections in humans [15]. Besides their roles as opportunistic pathogens, *Stenotrophomonas* sp., also have a strong genome-enabled propensity to biodegrade xenobiotic compounds (e.g., aromatics, heavy metals and biocides), as plant growth promoters and biological control agents [16,17]. Notably, previous genomic analysis performed on *Stenotrophomonas* sp., such as *S. maltophilia,* has revealed the presence of heavy metal resistance gene determinants against mercury with the presence of RTPA as well as copper and arsenic resistance [18,19]. In addition, Pages and coworkers showed that *S. maltophilia* was resistant to cadmium, selenium, and tellurium and to a lesser extent, other metals like cobalt, copper, zinc, nickel, mercury, silver, uranium, and lead [20].

Moreover, a growing body of research is continuing to demonstrate that the presence of heavy metals can co-select for antibiotic resistance in naturally-occurring environmental microbiota [21,22,23]. In this context, at least three different mechanisms are known in which co-selection occurs: (1) co-resistance, in which the metal resistance gene determinants are in close physical proximity with the antibiotic-resistant genes (also known as resistome), such as on the same mobilome (plasmid) or within the same cell (e.g., *merA* and *KPC* beta-lactamase; (2) cross-resistance, where the same resistome provides functions for efflux and antibiotic resistance, such as the *mdrL* which confers resistance to zinc, cobalt, and chromium as well as erythromycin, josamycin and clindamycin [24]; and (3) co-regulatory resistance, in which multiple genes, controlled by a single regulatory gene element, renders resistance to a host of toxic compounds, such as, antibiotics, biocides, and metals. The gene *czcR,* which regulates expression of the *CzcCBA* efflux pump, confers resistance to zinc, cadmium, cobalt along with antibiotic carbapenems [25].

In relation to the mechanisms that underpin heavy metal resistance and bioremediation, our ongoing studies continue to provide clues on the impacts of long-term exposure of native environmental microbiota to co-contaminants present in the impacted SRS soils [26,27]. Our ongoing studies unequivocally demonstrate that the SRS native microorganisms mount metabolic and stress responses against heavy metals such as uranium, nickel and mercury, using a suite of adaptive and evolutionary traits, e.g., the presence and by inference, activity of outer membrane efflux pumps, transporter proteins, stress/detoxification systems, cytochromes, and drug resistance determinants [26,27]. Here we report the genomic, physiological and cellular adaptations acquired by the newly isolated *Stenotrophomonas* sp., strain MA5 to survive in and colonize an Hg-impacted soil habitat. Furthermore, the strain was also found to be resistant against several antibiotics, indicating co-selection of mercury and antibiotic resistance is occurring within the SRS metal-impacted soils, which poses additional public health concerns, which should be addressed with further research.

## 2. Material and Methods

### 2.1. Isolation and Identification of Hg-Resistant Strain MA5

Strain MA5 was isolated from samples collected in the summer 2018 from the SRS Mills Branch location containing a total Hg concentration 10 ng/g (Xu et al., 2019, In Press). Soils were serially diluted on LB agar plates supplemented with 5 µg/mL of Hg and resulting colonies with different morphologies and color were further streaked and isolated for further work. One Hg-resistant strain, tentatively called MA5, was chosen for this study and was taxonomically identified by 16S rRNA gene sequencing as *Stenotrophomonas* sp.

### 2.2. Determination of Minimum Inhibitory Concentration (MIC) of Strain MA5 against Hg

Minimum inhibitory concentration against Hg was determined by screening growth at multiple levels of Hg using the Bioscreen C system (Growth Curves USA, Piscataway, NJ), as reported previously [27], with some minor modifications. The inoculum was prepared by growing the strain overnight at 30°C and diluting to an OD_600_ of 0.2. The assay was run using 30 µL of inoculum and 270 µL of LB medium, supplemented with Hg ranging from 0 to 30 µg/mL. The OD_600_ was measured in increments of every 3 h for up to 48 h. The experiment was run in triplicate and the average values are reported.

### 2.3. Hg Depletion Assay 

To determine Hg depletion by strain MA5, a single colony was inoculated in LB medium overnight and then diluted into fresh LB to an OD_600_ of 0.15 to 0.2. This medium was divided into two separate flasks; one served as a control and another one was amended with Hg and vigorously shaken at 30 °C. Supernatant and pellet fractions were collected and stored at 4 °C. Mercury concentration was measured following EPA method 7473 (thermal decomposition, gold amalgamation, atomic absorption detection) using an automated direct mercury analyzer (Milestone DMA-80). The instrument was calibrated using NIST-traceable standards. Analytical batches included blanks, replicates, and standard reference materials (NIST Spinach 1570a, Mussel 2976, and BCR-60 Aquatic Plant; NRCC DOLT3 and DORM2) with a range of certified Hg concentrations for QA purposes.

### 2.4. Hg Volatilization Assay

Hg volatilization was performed using the non-radioactive X-ray film method as described by Nakamura and Nakahara [28]. Specifically, strain MA5 was grown overnight in LB broth and then diluted to an OD_600_ of 0.2. This suspension was then transferred to a microplate and was amended with 3 µg/mL Hg and 5 µg/mL Hg, respectively. The plate was covered with X-ray film and incubated for 24 h in the dark, after which the film was developed. A foggy area appearing on the film was indicative of Hg^0^ volatilization due to the reduction of Ag^+^ on the X-ray film. Sterile LB medium and heat-killed cells of strain MA5, alone or with Hg, were used as controls, respectively.

### 2.5. Cell Viability Assay

Strain MA5 cells were grown overnight and further inoculated into fresh media with different concentrations of Hg. The assay was performed using Live/Dead BacLight^TM^ bacterial viability kit L7007 (Thermo Fisher Scientific, Carlsbad, CA, USA) according to the manufacturer’s instructions. Sample visualization was performed using an EVOS M5000 microscope (Thermo Fisher Scientific, Carlsbad, CA, USA). Live and dead bacterial cells were distinguished using fluorescence activity. Cells stained with green were considered as live, whereas red colored cells were considered dead. Bacterial cell counts were obtained with the analysis package EVOS M5000.

### 2.6. Microscopy Imaging

For cell imaging, 1–2 µL of the cell culture was imaged using differential interference contrast (DIC) and fluorescence techniques using an inverted EVOS M5000 microscope (Thermo Fisher Scientific, Carlsbad, CA, USA) with a 100x objective lens (1.3 NA).

To assess the cell morphology as a function of Hg treatment, cultures grown overnight were diluted 1:100 into fresh LB media supplemented with different Hg concentrations and grown at 30°C for 12 h. Samples were then collected, and several images were captured for each set of Hg treatment. Cellular lengths were measured in micrometers using the image analysis package EVOS M5000. For cell membrane visualization, a culture portion was stained with 5 µg/mL FM^TM^ 1-43FX (Thermo Fisher Scientific) for 10 min at room temperature and images were acquired using a GFP (Green fluorescent protein) filter set.

### 2.7. Antibiotic Susceptibility Test

The antibiotics susceptibility test was performed using the disc diffusion method [29]. For this, the culture was grown overnight in Muller Hinton broth and incubated at 30 °C. The cell suspension was then spread homogeneously over the MH agar plates and the appropriate antibiotic discs were placed on top (BD BBL^TM^ Sensi-DiscTM Antimicrobial Susceptibility Test Discs). Plates were incubated for 24 h at 30 °C and the zone of inhibition was recorded and interpreted according to the manufacturer’s instructions.

### 2.8. Genomic Characterization

This was performed as described in our recent reports [26,27]. Briefly, Genomic DNA from strain MA5 was extracted and prepared for sequencing on an Illumina HiSeq2000 instrument. Genome de novo assembly was performed on the obtained genomic contigs using the CLC genomics workbench (v11.0.1; Qiagen, Aarhus, Denmark), and sequences were trimmed using a quality threshold of Q20, and a requirement of 50 bases after trimming. The genome, with an average coverage of 200x, was then annotated and gene prediction was performed by RAST and NCBI’s Prokaryotic Genomes Automatic Annotation Pipeline (PGAAP), version 2.0. Amplicon-based Analysis of the mer Operon in Strain MA5

The presence of conserved region of *merA* was investigated through PCR amplification. The amplification was performed with the primers forward A1s-n.F (TCCGCAAGTNGCVACBGTNGG) and reverse A5-n.R (ACCATCGTCAGRTARGGRAAVA) as described by [30]. For identification purposes, PCR products were obtained using 27F and 1492R 16S rRNA gene primers, which is a standard technique.

### 2.9. Evaluation of the mer Operon in Strain MA5 Using RT-qPCR

A single bacterial colony was inoculated in LB medium and grown overnight and then diluted in fresh LB medium to an O.D._600_ of 0.15. This medium was divided into two separate flasks; one served as a control and another one was amended with 5 µg/mL of Hg. Cultures were vigorously shaken at 200 rpm and at 30 °C. Once OD_600_ reached 0.9, 25 mL of sample was collected and centrifuged at 8000 rpm for 10 min at 4 °C. The supernatant was discarded, and the pellet was washed once with cold sterile water, quickly frozen with liquid nitrogen and stored at −80 °C. Further, sample processing took place at a DNA sequencing facility, at the University of Illinois.

Briefly, RNA was extracted from cell pellets using the Maxwell^®^ RSC simply RNA Cells Kit (Promega, Madison, WI, USA), according to manufacturer’s instructions. The isolated RNA was reverse transcribed using a High-Capacity cDNA Reverse Transcription Kit with random primers (Applied Biosystems, Foster City, CA, USA). The cDNA synthesis was performed according to the manufacturer’s instructions. After assay efficiency and optimization, all samples were assayed using qPCR in triplicate in 10 µL reaction volumes. Each reaction contained 2.5 µL of cDNA, 5 µL of 2X TaqMan Fast Advanced Master Mix (Applied Biosystems), 0.5 µL of custom designed 20X IDT Gene Expression Assay with specific minor groove binding probe labeled with the 5′ reporter dye 6-carboxy-fluorescein (FAM). Amplification and detection were performed with a ViiA7 Real-Time PCR System (Applied Biosystems), under the following conditions: 2 min at 50 °C, 2 min at 95 °C, and 40 cycles of 1 s at 95 °C and 20 s at 60 °C. The reporter dye signal was measured relative to the internal reference dye (ROX) to normalize for non-PCR-related fluorescence fluctuations occurring from well to well. The automatic threshold cycle number generated after each run was used for all analyses.

The relative gene expression was calculated using the 2^-ΔΔCt^ method with each transcript signal normalized to 16S rDNA. Transcript signals for each treatment were compared to the transcript signal from the control group. Specific primers for the genes within *mer* operon and 16S rDNA were designed and shown in Appendix A. All runs were done in three biological replicates and their average value are shown with error bars representing one standard error. Differences in expression between controls and treatments were tested for significance by ANOVA using JMP 14 (SAS Inc.,Cary, NC, USA).

## 3. Results and Discussion

### 3.1. Hg depletion Potential of Stenotrophomonas sp. Strain MA5

An Hg-resistant strain, taxonomically closest to *Stenotrophomonas maltophilia*, was obtained from the Mills Branch location of the Savannah River Site (SRS), as part of a larger ongoing study. Taxonomic affiliation of strain MA5 using the one codex pipeline revealed tight clustering with *Stenotrophomonas* sp., as shown in Appendix A. To determine the resistance potential, different concentrations of Hg were supplemented in LB media and growth progression was assessed, which revealed that MA5 was able to grow with up to 20 µg/mL of Hg (Figure 1), albeit with an increased lag phase. Rapid growth was observed with up to 10 µg/mL of Hg and stationary phase reached in 24 h (Figure 1).

The Hg-resistance capability was further confirmed by Hg depletion analysis where microcosms were established with Hg followed by quantification of Hg depletion over the period of bacterial growth. This analysis showed that the initial concentration of Hg declined drastically in the supernatant by mid-log phase (~0.7 OD) (Figure 2A), suggesting that MA5 has the ability of Hg biosorption and/or bioaccumulation. Moreover, Hg quantification in the pellet fraction showed that an increase in Hg concentration by mid-log phase was followed by a rapid decline by the end of stationary phase or 24 h (Figure 2B), suggesting the possibility of Hg bioremediation via the volatilization process.

It is known that the Hg resistance system of bacteria is governed by *mer* operon and based on Hg^2+^ reduction into volatile Hg^0^ that then, diffuses out of the cell. Therefore, we evaluated Hg volatilization by MA5 grown in the presence and absence of Hg. This revealed the strong volatilization ability of strain MA5, as demonstrated by appearance of foggy areas on the X-ray film, whereas, no volatilization was detected from heat-killed cells of strain MA5 (Figure 2C). This result indicated that the strain may carry the *mer* determinants which could reduce and volatilize Hg, as also shown previously for *Pseudomonas* strain, *Bacillus cereus* [31,32].

### 3.2. Genome Analysis Showed Genetic mer Determinants in Stenotrophomonas sp. MA5

From the above results, it appears that strain MA5 possesses Hg-resistance and can bioaccumulate Hg from the surrounding environment. Moreover, the occurrence of Hg volatilization raised the possibility of *mer* determinant in this strain. Therefore, we sought to understand the molecular basis of Hg resistance and identify the *mer* determinants by using whole genomic sequence analysis of strain MA5. Approximately 8.5 M paired reads (average length 118 bases) were obtained after trimming and quality control, which were employed for assembly. The non-scaffolded assembly generated 264 contigs, with an N_50_ of 64,365 bases, and a total size of 4,513,544 bases with an average coverage of 200X. as recently reported by our group, the RAST-based annotation revealed 3921 coding sequences and a G+C content of 66.2 for strain MA5 (Pathak et al., 2019, Under review). Furthermore, annotation resulted in the binning of approximately 48% of MA5′s genome under 1856 subsystems with main gene categories of (count in parenthesis): carbohydrate metabolism (269); cofactors, vitamins, prosthetic groups, pigments (229); membrane transport (195); resistance to antibiotics and toxic compounds (130); and stress response (119). Several gene determinants known to encode resistance against heavy metals, including the cobalt-zinc-cadmium efflux system, the arsenic detoxification system, the chromate-inducible chrBACF operon, along with multiple membrane transporters were also identified, which potentially render ecologically and environmentally relevant soil survival traits for strain MA5.

Mining of the whole genome sequence of MA5 to assess mercury resistance mechanisms unequivocally showed the presence of a *mer* operon, consisting of *merR*, *merT*, *merP*, *merA, merD* and *merE* (Figure 3A); these ORF coordinates and their putative functions have been described in Table 1. Interestingly, we observed the presence of an ORF designated as a Tn21 protein of unknown function, Urf2, which was located next to the *merE* ORF, followed by two mobile genetic elements (Figure 3A), suggesting that the *mer* operon in strain MA5 may have a transposon-based origin [8,9]. Based on data obtained from whole genome sequence analysis, we propose that strain MA5′s mercury resistance is narrow-spectrum, as evidenced by the presence of the above-stated *mer* determinants but absence of *merB*, which confers broad-spectrum Hg resistance.

As we mentioned previously that MerA plays a central role in *mer* operon by converting Hg^2+^ into volatile Hg^0^. A previous study on taxonomic assessment of *merA* genes from Gram-negative bacteria divulged a highly conserved region at the C-terminus of the resulting protein (Figure 3B), carrying an oxidoreductase dimerization domain from Tn*501* MerA [30]. Based on this, Chadhain et al. designed a pair of degenerate PCR primers, A1s-n.F and A5-n.R (sequence shown in material and methods section), targeting a 285 bp fragment from the conserved region of *merA*. As expected, strain MA5 also harbored the conserved *merA* as shown by PCR amplification (Figure 3C). Furthermore, genome-centric assessment of *mer* determinants identified in strain MA5 revealed the presence of several gene homologues previously demonstrated to play a role in Hg-resistance. For example, homologues of *merA* in MA5 relative to 5 similar gene systems are shown in Figure 3D. Similarly, *merR, P, T, D,* and *E* are also shown with other homologous gene determinants in Figure 3E–I.

### 3.3. Increased Transcription of Mer Proteins in the Presence of Hg Stress

The above results provided evidence of *mer* gene determinants of *Stenotrophomonas* sp. strain MA5 conferring Hg-resistance for survival under Hg stress. Further, to elucidate the functional activity of *mer* operon in strain MA5, we evaluated the transcription of each gene of the *mer* operon using a qRT-PCR approach. Note that typically, mercury regulation by bacterial cells are performed by two genes, *merR* and *merD*. Specifically, the *merR*, a regulatory gene, encodes a metalloregulatory protein and is an Hg^2+^ dependent transcriptional repressor-activator such that it senses the metal concentration and controls the expression of the other functional *mer* operon genes. It is well known to bind with the promoter-operator region (*merOP*) and regulates *mer* operon gene expression in Hg-replete or Hg-depleted conditions. Hg-free-MerR then undergoes conformational structure changes upon binding to Hg^2+^ and switches from repression to the activation state Hg-MerR. The *merD* gene, present downstream to the *merA*, encodes a secondary regulatory protein and is expressed in very small amounts downregulating the *mer* operon. Its absence causes an increase in *mer* operon expression and it binds to the same operator-promoter region (*merOP*) as MerR, acting as an activation antagonist for MerR function.

With this background available, the functions of these mercury-cycling genes were investigated in strain MA5. Upon Hg treatment, transcription of *merT, P, A, D,* and *E* genes were found > 40 fold higher as compared to cells without Hg treatment (Figure 4A; treatments were significantly greater than controls by ANOVA, *p* < 0.001). In contrast, the transcription of *merR* was significantly higher (~1-fold, by ANOVA, *p* < 0.05) as compared to the control (Figure 4B). It has been shown that *merR* codes for a transcriptional regulator protein which ensures that the operon is transcribed only in the presence of Hg and hence MerR displays operonic regulation in both a negative and positive fashion [33]. Taken together, we propose that in the absence of Hg, the regulator protein MerR turns off the transcription of *merT, P, A, D,* and *E* (Figure 4C), and turns on expression only when the cellular environment senses the presence of Hg in the surrounding media (Figure 4D); these findings mirror earlier reports [34,35,36], but this is the first such report on mercury volatilization and molecular understanding of *mer* operon in a *Stenotrophomonas* sp.

### 3.4. Broad Antibiotic Resistance in Strain Stenotrophomonas sp. MA5

Due to their mobility, transposons can confer both, antibiotic and metal resistance to environmental microbiomes. Moreover, heavy metal resistance can also co-select for antibiotic resistance to the host bacterial cell [21,22,23,37]. Because our study suggests that the *mer* operon of strain MA5 is a part of a transposon, we determined the resistivity of strain MA5 against various antibiotics, and found resistance against ampicillin, rifampicin, nitrofurantoin, erythromycin, tetracycline, streptomycin, and penicillin. Intermediate resistance was also found against nalidixic acid and sensitivity was recorded against chloramphenicol, sulfamethoxazole, and kanamycin as shown in Table 2. This suggests that strain MA5 possesses both, multiple antibiotic as well as mercury resistance. Note that it has been demonstrated that if the antibiotic and metal resistance encoding genes are physically present on the same genetic element or plasmid, they become co-selected during the horizontal gene transfer processes [38]. Because Hg resistance and antibiotics genes are commonly found on the same plasmid or transposon(s); thus, exposure to mercury also promotes survival and colonization of bacteria resistant to multiple antibiotics, despite the absence of antibiotic selective pressure. In many cases, HgR bacteria have been shown to carry antibiotic resistance towards ampicillin, tetracyclin, erythromycin, and penicillin [39]. Moreover, HgR bacteria are resistant to multiple antibiotics relative to Hg sensitive (HgS) bacteria. This phenomenon presents a problem as heavy metal contaminated sites can become an evolutionary repository source for multiple antibiotic resistance to develop resulting in public health concerns.

### 3.5. Cell Forms a Filamentous Structure as a Result of Mercury Stress

It is well-known that mercury exerts toxic effects on all life forms including microorganisms. However, genomic plasticity and morphological flexibility enable microbiota to survive in the presence of Hg. To study the physiological response, when exposed to Hg stress, we first evaluated bacterial cell morphology. Interestingly, we noticed that cell size increased as a function of increasing Hg concentration (Figure 5A–F). The average cell size of *Stenotrophomonas* sp. MA5 was estimated to be ~4.8 µm which increased to as much as 12.3 µm upon exposure of 20 µg/mL Hg (Figure 5E,F). Moreover, staining of the bacterial cell membrane using FM 1-43FX confirmed this observation as shown in Figure 6A–E. This is indicative that MA5 cells continue to grow but cease to divide into daughter cells due to Hg stress. This phenotype is related to the level of Ftsz, a cell division protein, which decreases and therefore daughter cells do not separate [40]. Filamentation has also been reported to occur by SOS response induced by DNA damage [41,42]. It is noteworthy that microorganisms have been shown to elongate their cell size to achieve survival upon exposure to different kinds of stressors [43]. It is tempting to hypothesize that multinucleated filamentous structure increases the probability of recombination between chromosomes. During this process, mutant alleles, generated by SOS response, may get preferentially selected and thus potentially resulting in an ecologically “evolved” generation that can survive better under conditions of environmental stress [44].

Taken together, the above results suggest that strain MA5 possesses high Hg-resistance due to the *mer* operon and its ability to biotransform Hg^2+^. However, increased Hg concentration did impose cellular stress, and the strain responds by cellular elongation. Furthermore, we also tested the effect of an increasing Hg concentration on cell viability. The cell viability assay showed ~95% cells were viable even at 20 µg/mL of Hg (Figure 6F–I), indicating that elongated cells did not lose much cell viability but became deprived of the ability to divide. Interestingly, while performing the cell viability assays, we noticed the presence of segregated nucleoids, as revealed by chromosomal stain Syto 9 (Figure 6H). This suggests that Hg, at tested concentrations, did not interfere with nucleoid division but affected cell division by inhibiting daughter cell separation or cytokinesis. It would be interesting to study whether Hg inhibits the Ftsz synthesis or any other factors which contribute towards the cell division process. Specific mechanism(s) resulting in were in the change of bacterial cell morphology as a function of Hg stress remains to be fully understood at this time. Overall, this study provides further understanding on genomics and physiological adaptations of *Stenotrophomonas* sp. MA5, for survival in an Hg-contaminated habitat, which is of importance to Hg-contaminated environments across the globe.

## Figures and Tables

**Figure 1 cells-08-00309-f001:**
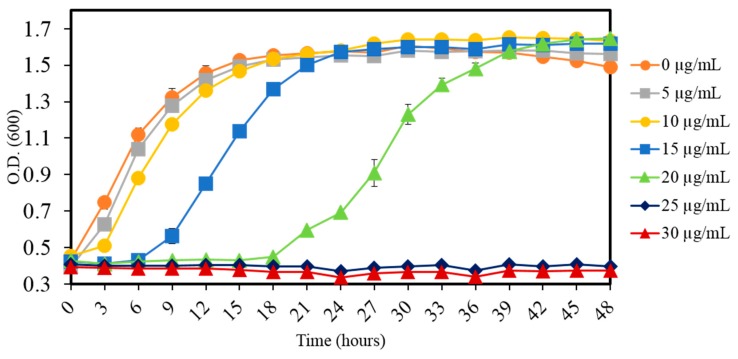
The growth profile of *Stenotrophomonas* sp. MA5 is shown, demonstrating resistance ability at different Hg concentration ranging from 0 to 30 µg/mL. The experiment was performed in triplicate and the values are shown in error bars depicting standard deviation.

**Figure 2 cells-08-00309-f002:**
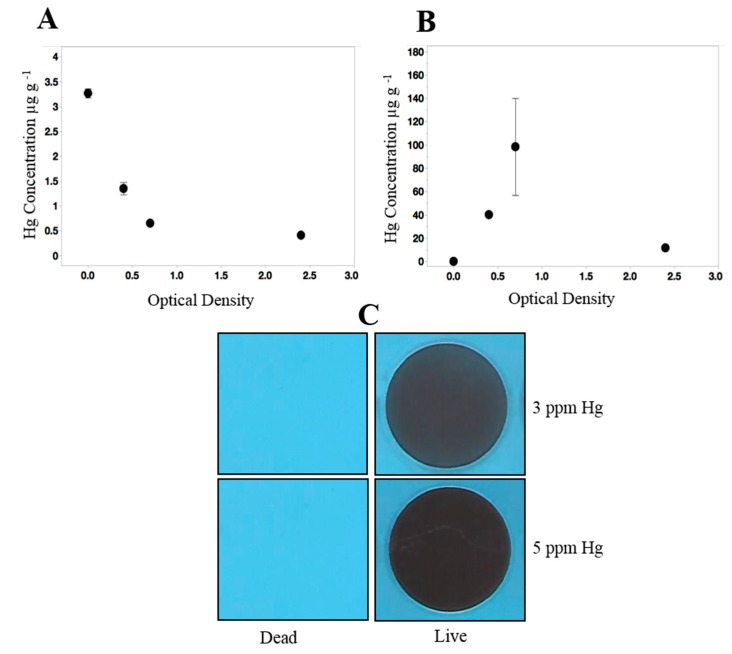
Hg removal potential of strain *Stenotrophomonas* sp. from the supernatant **(A)** and pellet **(B)** fractions is shown. **(C)** Hg volatilization assay from live and dead cells of *Stenotrophomonas* sp. treated with 3 and 5 µg/mL Hg. The experiment was performed in triplicate and representative images are shown here.

**Figure 3 cells-08-00309-f003:**
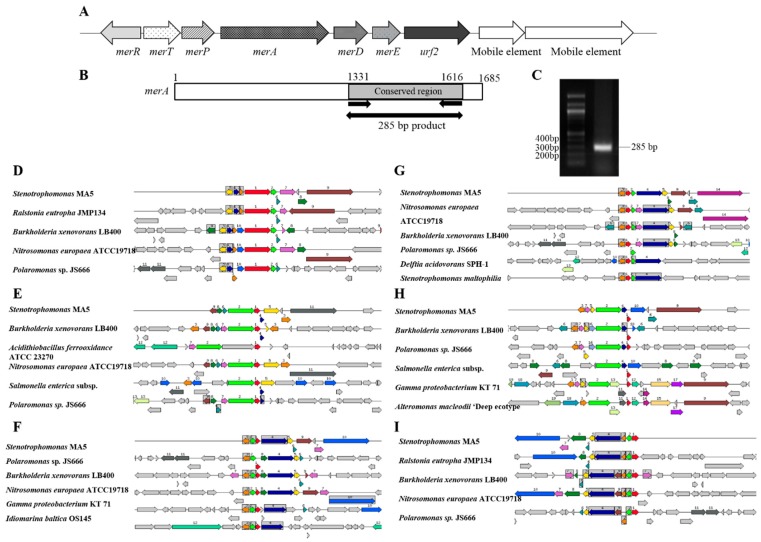
Presence of *mer* determinants in *Stenotrophomonas* sp. strain MA5 (**A**) A schematic showing genes present in *mer* operon of MA5 and their orientation on the chromosome. (**B**) PCR amplification of a conserved region of *merA* using primers A1s-n.F and A5-n.R and presence of the amplified band on the agarose gel (**C**). (**D**–**I**) Shown are the chromosomal regions of *mer* operon genes in *Stenotrophomonas* sp. strain relative to 5 other organisms. The graphic is centered on the focus genes *merA*, *merD, merP, merT*, *merE* and *merR* respectively, red in color and numbered 1. Sets of genes with a similar sequence are grouped with the same number and color. Genes whose relative position is conserved in other species are functionally coupled and share gray background boxes.

**Figure 4 cells-08-00309-f004:**
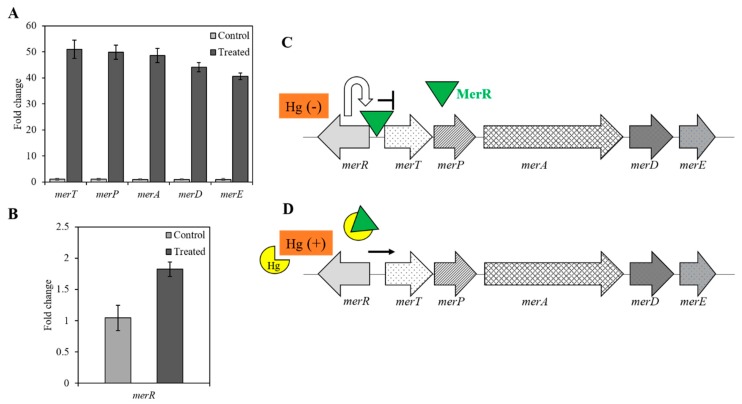
The transcription profile of *mer* determinants upon Hg exposure in strain MA5. Graphs show fold change value obtained after qRT-PCR analysis of and *merT, merP, merA, merD, and merE* (**A**) *merR* (**B**) in the absence and presence of 5 µg/mL Hg. The assay was performed in triplicate and values are shown with error bars depicting standard deviation. P values were calculated using one-way ANOVA analysis to calculate the significant difference between control v/s treated cells. The P value was < 0.001 for *merT, P, A, D,* and *E,* and < 0.03 for *merR*. Schematic showing MerR mediated regulation of *mer* operon in the absence (**C**) and the presence of Hg (**D**).

**Figure 5 cells-08-00309-f005:**
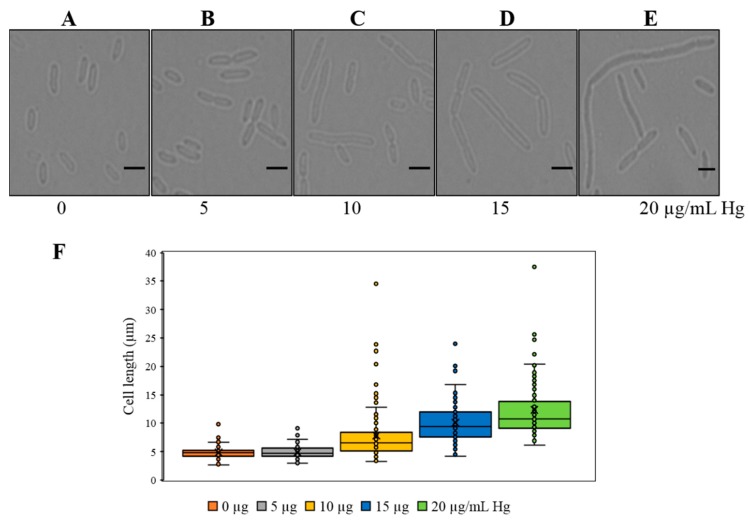
Effect of different Hg concentrations on cell morphology of strain *Stenotrophomonas* sp. MA5 (**A**–**E**) representative microscopic images of cells treated with 0 to 20 µg/mL Hg respectively. Scale bar = 5 µm. (**F**) Box plot showing cell lengths measurements. 100 cells were analyzed for each Hg concentration. P values were calculated using one-way ANOVA analysis to calculate the significant difference between control v/s treated cells. For the cells treated with 5 µg/mL Hg was not found significantly different than 0 µg/mL Hg and P value was <1.0. Whereas, cells with 10, 15, 20 µg/mL Hg were found significantly different than 0 µg/mL Hg with a *p* value of <0.0001.

**Figure 6 cells-08-00309-f006:**
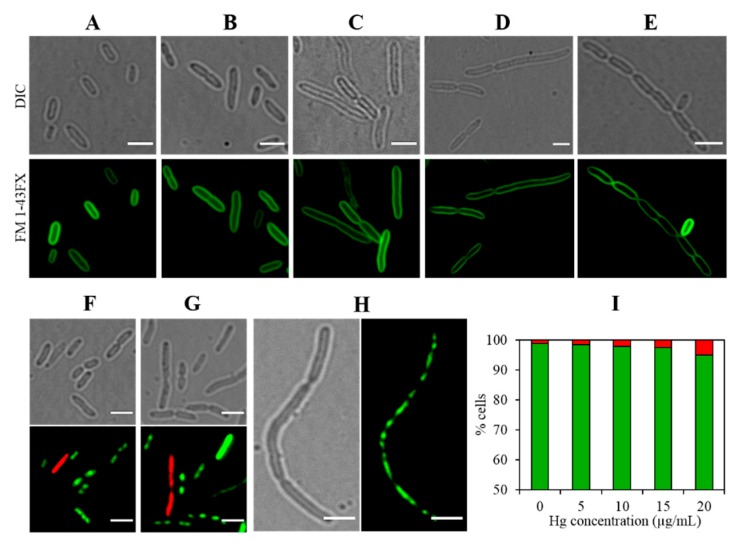
The cell viability assay performed on *Stenotrophomonas* sp. MA5 grown in the presence of Hg at concentrations ranging from 0 to 20 µg/mL. (**A**–**E**) Representative fluorescence micrographs of cells stained with membrane marker FM 1-43FX. Scale bar = 5 µm. Note that a larger area is shown in images D and E to cover the whole filamentous structure from a single cell. (**F**–**H**) Representative images for cell viability assay. Cells stained with green represent live and stained with red represent dead cells. Scale bar = 5 µm. (I) The histogram shows the percentage of live cells (green in color) and dead cells (red in color), at each tested Hg concentration. More than 300 cells were observed for each treatment.

**Table 1 cells-08-00309-t001:** Genes identified for mercury resistance in *Stenotrophomonas* sp. strain MA5.

Feature	Coordinates	Length (bp)	Putative Role
*merR*	57147-57581	434	Regulatory protein
*merT*	57075-56725	350	Mercuric transport protein
*merP*	56712-56437	275	Periplasmic mercuric ion binding protein
*merA*	56365-54680	1685	Mercuric ion reductase
*merD*	54662-54297	365	Co-regulatory protein
*merE*	54300-54064	236	Transport protein

**Table 2 cells-08-00309-t002:** The list of antibiotics and susceptibility tested for *Stenotrophomonas* sp. strain MA5.

Antibiotics	Disc Concentration (µg)	Antibiotics Susceptibility
Penicillin	10	Resistant
Streptomycin	10	Resistant
Tetracycline	30	Resistant
Erythromycin	15	Resistant
Chloramphenicol	30	Susceptible
Nitrofurantoin	300	Resistant
Sulfamethoxazole	23.75	Susceptible
Kanamycin	30	Susceptible
Nalidixic acid	30	Intermediate
Ampicillin	10	Resistant
Rifampicin	30	Resistant

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
