# Peer review of "Multiple Lines of Evidences Reveal Mechanisms Underpinning Mercury Resistance and Volatilization by Stenotrophomonas sp. MA5 Isolated from the Savannah River Site (SRS), USA"

_cells, 2019, doi:10.3390/cells8040309_

Round 1
Reviewer 1 Report
Journal: Cells
Manuscript number: cells-467512
Article type: Article
Title: Multiple Lines of Evidences Reveal Mechanisms Underpinning Mercury Resistance and Volatilization by Stenotrophomonas sp. MA5 isolated from the Savannah River Site (SRS)
This paper aimed to reveal mechanisms underpinning mercury resistance and volatilization by Stenotrophomonas sp. MA5. The results seem interesting. However, the correct format and pretty language are important for article published. Therefore, I recommend this paper can be accepted after minor revision. Specific suggestions are provided below.Specific comments:
1. In the parts of ‘conclusion’, authors should make the article more visualized, accurate, and rigorous.
2. The authors are suggested to reword figure captions.
3. Statistical analysis on data in Figure 4 was not carried out using ANOVA. I suggest the authors to add the statistical analysis.
4. Some grammatical mistakes in the whole paper should be carefully corrected, tenses of sentences need to be consistent and corrected.
5. Some language expressions should be polished during the revision. Please improve the language greatly.
6. Please pay attention to the writing of superscripts and subscripts of letters/ numbers and format error in the whole manuscript, including reference.
7. I recommend that authors add some newly references concerning the cytotoxicity effects of toxic pollutants, such as Journal of Hazardous Materials 321 (2017) 37-46, Chemosphere 196 (2018) 575-584, Chemosphere 203 (2018) 199-208, Chemosphere 211 (2018) 573-583, and Chemosphere 224 (2019) 554-561.
Author Response
Following is our response. Shown in black color are the reviewer's comments and the author’s responses appear in red color.
Reviewer 1:
In the parts of ‘conclusion’, authors should make the article more
visualized, accurate, and rigorous.
Author’s response: The article has been revised as per suggestion. Changes were made in track change mode.
2. The authors are suggested to reword figure captions.
Author’s response: All figure captions have now been reworded.
3. Statistical analysis on data in Figure 4 was not carried out using ANOVA. I
suggest the authors to add the statistical analysis.
Author’s response: Statistical analysis has now been done and added in the figure legends. The figure was redrawn, and control v/s treated samples were put together to show the statistical analysis and their statistical difference.
4. Some grammatical mistakes in the whole paper should be carefully
corrected, tenses of sentences need to be consistent and corrected.
Author’s response: The article has been revised as per suggestion. Changes made can be easily tracked in track change format.
5. Some language expressions should be polished during the revision.
Please improve the language greatly.
Author’s response: The article has been revised as per suggestion. Changes made can be easily tracked in track change format.
6. Please pay attention to the writing of superscripts and subscripts of letters/
numbers and format error in the whole manuscript, including reference.
Author’s response: The article has been revised as per suggestion. Changes made can be easily tracked in track change format.
7. I recommend that authors add some newly references concerning the
cytotoxicity effects of toxic pollutants, such as Journal of Hazardous
Materials 321 (2017) 37-46, Chemosphere 196 (2018) 575-584,
Chemosphere 203 (2018) 199-208, Chemosphere 211 (2018) 573-583, and
Chemosphere 224 (2019) 554-561.
Author’s response: Thank for bringing these new articles to our notice. These articles pertain to the toxicity of silver nanoparticles to the fungi- Phanerochaete chrysosporium. Our work is on the mercury bioremediative mechanism by a newly isolated bacteria- Stenotrophomonas strain MA5, and therefore, unrelated to the suggested references. Therefore, we chose not to cite these works in our study.
Reviewer 2 Report
This is a very well written MS.
Some minor points:
Fig. 1 legend : do not start with the verb. i.e. The growth profile ... is shown. The values (plural)...
fig 2 legend, same as above
fig 4 legend, same as above
table 2 legend: delete "displays"
fig 6 legend, same as above
page 11, lines 12-14: can you split please this phrase to two phrases?
page 11, lines 23-24: it is not cleat what you mean, can you please rephrase?
Author Response
Following is our response. Shown in black color are the reviewer's comments and the author’s responses appear in red color.
We greatly appreciate your time in reviewing our manuscript so thoroughly which has facilitated improvement in this iteration of our study. We hope that this version will be acceptable for publication.
Fig. 1 legend: do not start with the verb. i.e. The growth profile ... is shown.
The values (plural)...
fig 2 legend, same as above
fig 4 legend, same as above
table 2 legend: delete "displays"
fig 6 legend, same as above
Author's Response: All the figure legends have been rephrased to not begin with a verb.
page 11, lines 12-14: can you split please this phrase to two phrases?
Author's Response: This phrase has been split.
page 11, lines 23-24: it is not cleat what you mean, can you please
rephrase?
Author's Response: This has been rephrased.